# Multiple Levels of Control Processes for Wisconsin Card Sorts: An Observational Study

**DOI:** 10.3390/brainsci9060141

**Published:** 2019-06-17

**Authors:** Bruno Kopp, Alexander Steinke, Malte Bertram, Thomas Skripuletz, Florian Lange

**Affiliations:** 1Department of Neurology, Hannover Medical School, Carl-Neuberg-Straße 1, 30625 Hannover, Germany; Steinke.Alexander@mh-hannover.de (A.S.); Malte.Bertram@stud.mh-hannover.de (M.B.); Skripuletz.Thomas@mh-hannover.de (T.S.); Florian.Lange@kuleuven.be (F.L.); 2Behavioral Engineering Research Group, KU Leuven, Naamsestraat 69, 3000 Leuven, Belgium

**Keywords:** Wisconsin Card Sorting Test, executive control, response-related learning, behavioral avoidance, error-suppression effect

## Abstract

We explored short-term behavioral plasticity on the Modified Wisconsin Card Sorting Test (M-WCST) by deriving novel error metrics by stratifying traditional set loss and perseverative errors. Separating the rule set and the response set allowed for the measurement of performance across four trial types, crossing rule set (i.e., maintain vs. switch) and response demand (i.e., repeat vs. alternate). Critically, these four trial types can be grouped based on trial-wise feedback on *t* − 1 trials. Rewarded (correct) maintain *t* − 1 trials should lead to error enhancement when the response demands shift from repeat to alternate. In contrast, punished (incorrect) *t* − 1 trials should lead to error suppression when the response demands shift from repeat to alternate. The results supported the error suppression prediction: An error suppression effect (ESE) was observed across numerous patient samples. Exploratory analyses show that the ESE did not share substantial portions of variance with traditional neuropsychological measures of executive functioning. They further point into the direction that striatal or limbic circuit neuropathology may be associated with enhanced ESE. These data suggest that punishment of the recently executed response induces behavioral avoidance, which is detectable as the ESE on the WCST. The assessment of the ESE might provide an index of response-related avoidance learning on the WCST.

## 1. Introduction

The Wisconsin card sorting task was originally introduced in the 1940s [1,2]. It has a long history of application in neurological patients, which dates back to the 1950s and 1960s [3,4]. Milner’s [3] observation that patients who suffer from large-scale excisions of their frontal lobes exhibited much stronger perseverative behavior (see the following paragraph than patients with posterior excisions contributed in a particular way to the success story of the Wisconsin card sorting task in clinical neuropsychology, finally paving the way for a number of standardized Wisconsin Card Sorting Test (WCST) variants [5,6,7,8]. Until recently, these WCST variants are still considered to be the gold standard for the neuropsychological assessment of executive functioning [9,10] (see [11,12,13,14,15,16] for reviews). 

The WCST consists of four stimulus cards, which were placed in front of the subject: They depict a red triangle, two green stars, three yellow crosses, and four blue circles, respectively. The subject receives two sets of 64 response cards, which can be categorized according to color, shape, and number. The subject is told to match each of the response cards to one of the four stimulus cards and is then given feedback on each trial whether he or she is right or wrong. The task requires establishing cognitive sets (e.g., to sort cards according to the abstract color dimension). It is necessary to maintain it in response to positive feedback once a cognitive set has been established. On the contrary, shifting the prevailing cognitive set is requested in response to negative feedback. The task provides information on several aspects of executive function beyond basic indices, such as task success or failure. Important indices of performance on the task include the number of categories that are achieved (i.e., the number of sequences of six or ten consecutive correct sorts; the actual length of the sequence depends on the test version), the number of perseverative errors (i.e., the number of failures to shift cognitive set in response to negative feedback), and the number of set-loss errors (i.e., the number of failures to maintain cognitive set in response to positive feedback). 

The WCST is perhaps the most frequently used neuropsychological test in the assessment of cognitive flexibility, an ability that involves shifting between cognitive sets as required by task demands [14,17,18]. However, despite its success, a number of problems remain as significant limitations to the WCST. The lack of a comprehensive theoretical framework for its performance measures, moderate reliability, largely unestablished validity [19], and complex scoring rules that do not easily lend themselves to comprehensible measurement are among these problems [20,21]. As a corollary of these problems, the WCST must be considered as a relatively impure assessment tool, which needs very cautious interpretation by the examiner [22,23,24,25,26]. 

The present study focused on the outlines of a task theory for the WCST to improve our understanding of the mental processes that underlie WCST performance. Successful performance on the WCST depends on the utilization of trial-by-trial positive feedback (‘this is correct’) and negative feedback (‘this is incorrect’) during test administration. WCST performance is typically considered to be categorical, that is, it is generally assumed that the examinees relate the feedback to the most recently applied sorting rule (e.g., matching cards according to color). To the degree that participants learn at this categorical level, they will apply the identical sorting rule on the upcoming trial when the feedback has been positive, but they will apply another sorting rule when the feedback has been negative (see Figure 1a). It is important to note that this cognitive task theory provides the conceptual basis for considering the WCST as a neuropsychological test that is suitable for the assessment of executive functions (i.e., control processes exerted at an abstract, categorical level). 

Nowadays, it is less common to consider the possibility that learning at the level of motor responses may also contribute to WCST performance. This is surprising given that predominant theoretical frameworks, both in neuropsychology (e.g., [27]) and in cognitive neuroscience (e.g., [28,29]), typically involve multiple, hierarchically organized levels of control processes. In fact, as early as 1965, Luria [30] already distinguished between cognitive perseveration (an inability to change an afferent readiness) and motor perseveration (an inability to change an efferent readiness). Response perseveration has also been considered in the context of task-switching research in cognitive psychology (see [31] for an overview). For example, Hübner and colleagues proposed that responses that have been executed on particular experimental trials receive inhibition to prevent their re-execution on subsequent trials (e.g., [32]). Motor control processes have also received attention in the cognitive neurosciences, where this level of control is typically attributed to lower-order fronto-striatal (more specifically, frontal-subthalamic) loops (e.g., [33]). In contrast, learning and control at the level of motor responses (i.e., the choice of particular stimulus cards), rather than at the categorical level (i.e., sorting according to one of the three abstract WCST rules), has so far been a largely neglected topic in the context of human performance on the WCST.

Performance on the WCST might not only be affected by categorical learning of the WCST rules, but also by learning processes that are related to the sorting responses via trial-by-trial feedback. Our study intends to detect the potential behavioral indices of response-related (pre-categorical) learning on the WCST. To our knowledge, this has never been attempted before. We analyzed WCST conditional error probabilities that were obtained from a relatively large sample of inpatients who suffered from diverse neurological conditions (*N* = 146) on the Modified WCST (M-WCST) to examine the contribution of response-related learning on WCST performance [8]. A new M-WCST error scoring method was developed, which combined the experimental factors *rule sequence* (requested rule maintenance/switch) and *response sequence* (requested response repetition/alternation) across consecutive trials. The requested rule sequence is indicated by the feedback that was obtained on the previous trial. A positive feedback stimulus (‘Correct’) indicates that the executed rule should be maintained, whereas a negative feedback stimulus (‘Incorrect’) indicates that a rule switch was required. 

Figure 1 illustrates the response-sequence factor. Consider, for example, a particular trial *t* − 1, on which the outside left card (i.e., one red triangle) was chosen. The analysis of the response sequence refers to the succession of requested card choices (i.e., card choices on trial *t* that are in accord with the type of feedback that occurred on trial *t* − 1) across two consecutive trials. Specifically, an error occurred on a response alternation sequence when the outside left card was chosen again in the subsequent trial *t*, although the participants should have altered their responses by choosing a different card than on trial *t* − 1. In other words, the participants erroneously repeated their card-selection response when committing an error in response-alternation sequences. An error occurred on a response repetition sequence when any of the remaining cards was chosen on trial *t*, although the participants should have repeated their responses by choosing the same card as on trial *t* − 1. In other words, participants erroneously alternated their card-selection response when committing an error in response-repetition sequences. Conditional probabilities of the occurrence of errors on the four distinguishable rule sequences/response sequences were analyzed (see Section 2. Materials and Methods for details). 

Figure 1a also highlights the novelty of the present analysis, which lies in the factorial design that allows for combining the requested rule sequence (as indicated via type of feedback, i.e., positive/negative feedback on trial *t* − 1) and response sequence. Importantly, this analysis allows for investigating the potential effects of response-specific learning on the WCST. This type of learning should affect conditional error probabilities in the case that response-related learning proves to be effective. The specific predictions that can be derived from this conceptualization of WCST performance are most easily understood when the response-alternation sequences are considered, as shown in Figure 1b. In these trials, an error-enhancement effect (EEE) driven by reward might occur, as indicated by the black arrow on one of the potentially wrong card choices. In contrast, an error-suppression effect (ESE) that was driven by punishment might occur, as indicated by the black cul-de-sac sign on the wrong card choice.

## 2. Materials and Methods

### 2.1. Participants

The initial sample consisted of 146 (58 female) inpatients who suffered from diverse neurological conditions and who were consecutively referred to an experienced neuropsychologist (BK) for a neuropsychological evaluation. Diagnostic assignments were conducted by an experienced neurologist (TS) who was blinded from the neuropsychological characteristics of individual patients with regard to the effects of interest. The diagnostic assignments were atypical Parkinson’s disease (progressive supranuclear palsy (PSP), cortico-basal degeneration (CBD), multi-system atrophy-Parkinsonian subtype (MSA-P); *n* = 25), (early) Alzheimer’s disease/mild cognitive impairment (*n* = 14), frontotemporal lobar degeneration (*n* = 26), vascular encephalopathy (*n* = 10), stroke (*n* = 13), multiple sclerosis (*n* = 14), normal pressure hydrocephalus (*n* = 14), depression (*n* = 13), neuropathy (*n* = 10), cognitive impairment with unknown origin (*n* = 4), and no neurological (or psychiatric) disease (*n* = 3). The study received institutional ethics approval (Ethikkommission at the Hannover Medical School; ID 7529, 5.7.2017) and it was in accordance with the 1964 Helsinki Declaration and its later amendments. Written informed consent was obtained from participants. Table 1 summarizes the sociodemographic and neuropsychological characteristics of the sample, divided into subsamples of 112 patients who could be included in the final confirmatory analyses, and 34 patients who had to be excluded from these analyses due to missing data, as detailed below.

### 2.2. Materials and Design

All of the patients performed the M-WCST [8] and the extended German version of the Consortium to Establish a Registry for Alzheimer’s Disease Neuropsychological Assessment Battery (CERAD-NAB) [34,35,36,37]. 

The M-WCST represents a well-established, commercially available variant of Wisconsin Card Sorting Tests. In contrast to the standard M-WCST instructions, a fixed order of task rules was pretended to acquire, namely {color, shape, number, color, shape, number} throughout this study. The major innovation of this study was the derivation of novel M-WCST error scores, as illustrated in Figure 1a and as detailed below.

The novel M-WCST scores incorporate the scoring of the traditional set-loss and perseveration errors, which both represent a major outcome variable in many WCST-studies. Here, we refer to these error types by the requested rule sequence, which is either a rule maintenance or rule switch with corresponding set-loss and perseveration errors, respectively. A set-loss error was committed by erroneously switching the rule when a rule maintenance was requested (i.e., indicated by a positive feedback on trial *t* − 1). For example, as illustrated in Figure 1a (left panels), the Shape rule was correct on trial *t* − 1, as indicated by positive feedback and the participant is requested to maintain the Shape rule on trial *t*. However, a set-loss error is committed if the participant erroneously switches to either Color or Number. A perseveration error was committed by erroneously maintaining the rule when a rule switch was requested (i.e., indicated by a negative feedback on trial *t* − 1). In Figure 1a (right panels), the participant received negative feedback for the Number rule on trial *t* − 1, which requests a rule switch. The erroneous maintenance of the Number rule on trial *t* is scored as a perseveration error. Please note, that, on trials that request a rule switch, both correct responses include the actual correct response as well as a so-called ‘efficient’ error [38,39] on trial *t*. This holds true because the M-WCST utilizes implicit (‘transition’) task cues [8] and three task rules (i.e., Color, Shape, Number), thereby evoking ambiguity with regard to the correct task rule after the occurrence of negative (‘incorrect’) feedback on trial *t* − 1, which rendered it a matter of chance whether or not the correct rule is executed on trial *t* [14,40,41,42].

The novelty of our error scoring is the factorial design that allows for differentiating set-loss and perseveration errors by a requested response repetition or a requested response alternation (factor response sequence, Figure 1a). A trial was classified as a requested response repetition if the correct response (or correct responses for rule switch trials) included the given response on trial *t* − 1. A trial was classified as a requested response alternation if the correct response did not include the given response on trial *t* − 1. The factors rule sequence and response sequence allow for comparing performance on trials with a requested repetition or alternation of a rewarded response (requested rule maintenance/requested response repetition; and, requested rule maintenance/requested response alternation, respectively) and trials with a requested repetition or alternation of a punished response (requested rule switch/requested response repetition; and, requested rule switch/requested response alternation, respectively). Thereby, we introduce a novel error scoring to test for short-term behavioral plasticity on the WCST.

Conditional error probabilities were computed by dividing the number of committed errors by the sum of the committed errors and correct responses. We excluded the rarely occurring ‘odd’ errors on trial *t* from consideration (0.02% of all trials). An ‘odd’ error was committed when the stimulus card that matches no valid task rule was chosen (e.g., the far right stimulus card on trial *t* in Figure 1a). Likewise, the trials that followed a shift of the correct rule (i.e., the first trial on a new trial run) were excluded from consideration.

Table 2 shows the mean numbers of occurrence (and inter-individual variability) of each of these distinguishable types of errors. Note that the perseveration errors with a requested response repetition strongly outnumbered the remaining types of errors. 

Table 3 contains a short description of the extended CERAD-NAB. We computed a composite CERAD-NAB score [43], which integrates *Animal fluency* (max. = 24 by truncation), *Abbreviated Boston Naming Test* (max. = 15), *Word list learning* (max. = 30), *Constructional praxis* (max. = 11), *Word list recall* (max. = 10), and *Word list Discriminability* (max. = 10). However, we did not score *Word list discriminability* as the number of true positives minus the number of false positives, as in the original publication [43]. Instead, we defined *Word list discriminability* as a percentage (max. = 100 percent), being similar to the definition that was provided by the CERAD-NAB [34]. In order to put *Word list discriminability* on a comparable scale, as required by the composite score (i.e., max = 10), we divided the *Word list discriminability* score by 10, resulting in the formal definition that is displayed in Table 3. We refer to this composite score as the *Adjusted Chandler score*. Its intention is to provide an approximate index of dementia severity. 

Missing or inappropriate data led to the exclusion of 34 patients (nine female) from the confirmatory analyses, resulting in a final sample of 112 patients (49 female). We had to exclude 18 patients from analysis who were unable to complete the M-WCST. We further excluded 16 patients from the analysis who did not have the opportunity to commit all of the above-defined types of error. This could happen for a number of reasons. For example, a patient who incorrectly reiterated one particular cognitive set (e.g., number) throughout all 48 trials would have no opportunity to commit set-loss errors simply because he or she would never be requested to maintain the rule, i.e. receives correct feedback. Table 1 contains a detailed description of the included and excluded patients’ sociodemographic and neuropsychological characteristics. An inspection of Table 1 reveals that excluded patients who completed the M-WCST achieved similar numbers of M-WCST categories (*M* = 3.19, *SD* = 2.23) when compared to the included patients (*M* = 2.99, *SD* = 1.87). 

### 2.3. Statistical Analyses

The newly derived M-WCST conditional error probabilities (see Section 2.2. Materials and Design and Figure 1a) were subjected to a two-way repeated measures analysis of variance (ANOVA) with the factors rule sequence (requested rule maintenance/switch) and the response sequence (requested response repetition/alternation). The level of significance was set to *p* < 0.05. We also conducted a Bayesian ANOVA to quantify evidence in favor of each model that was considered in the ANOVA in addition to traditional null hypothesis significance testing [44]. All of the analyses were computed while using JASP version 0.8.5.1 [45]. Default settings of JASP were used for the Bayesian ANOVA. In addition to posterior probabilities, we reported logarithmized Bayes factors (logBF), which quantify the support for a hypothesis over another. For example, logBF_10_ = 2.30 indicates that the alternative hypothesis is approximately ten times more likely than the null hypothesis, as the corresponding Bayes factor is exp(logBF_10_) = 10.

We also ran correlation analyses to examine whether the newly derived M-WCST conditional error probabilities can be dissociated with regard to their relationships to traditional neuropsychological measures. In a first step, we computed the Spearman correlation coefficients for the relationships between perseveration errors with a requested response repetition and alternation, as defined in Figure 1a, and the subtest scores of the CERAD-NAB, age, and years of education. We then tested whether any of those variables was differentially related to the perseveration errors with a requested response repetition in comparison to perseveration errors with a requested response alternation, thereby indicating an association between CERAD-NAB subtest scores or demographic variables and the individual strength of the ESE. To this end, the corresponding correlation coefficients were compared according to the procedure that was outlined by Meng, Rosenthal, and Rubin [46].

Finally, we conducted in a first step a descriptive analysis of the strength of the ESE across the various diagnostic entities that were present in the current sample of consecutively referred neurological patients (see Figure 3). Subsequently, we tested whether apparent differences between diagnostic entities achieved conventional levels of statistical significance in two-way mixed model ANOVA and Bayesian ANOVA with the between-subjects factor diagnostic entity and the within-subjects factor response sequence (requested response repetition/alternation) with perseveration errors with a requested response repetition or alternation entering these analyses. 

## 3. Results

### 3.1. Confirmatory Analyses

Figure 2 shows the observed conditional error probabilities, separately for the four distinguishable rule sequences/response sequences. The two-way repeated measures ANOVA with the factors rule sequence (requested rule maintenance/switch) and the response sequence (requested response repetition/alternation) revealed the statistically significant main effects of rule sequence, *F*(1,111) = 27.00, *p* < 0.001, $\eta_{p}^{2}$ = 0.20, and response sequence, *F*(1,111) = 22.06, *p* < 0.001, $\eta_{p}^{2}$ = 0.17, and a statistically significant interaction between rule sequence and response sequence, *F*(1,111) = 10.09, *p* = 0.002, $\eta_{p}^{2}$ = 0.08. The Bayesian repeated measures ANOVA compared the likelihoods of the data under each model regarded in the ANOVA to a null model that included neither the main effects nor the interaction. The results indicated that the observed data were most likely under the model that included both main effects and the interaction (posterior probability = 0.911, logBF_10_ = 21.23), followed by the model that included both the main effects (posterior probability = 0.089, logBF_10_ = 18.90), and the models that included single main effects (posterior probability < 0.001, logBF_10_ = 13.10; and, posterior probability < 0.001, logBF_10_ = 5.13, for the model including rule sequence and the model including response sequence, respectively). A direct comparison of the models revealed that the observed data were ten times more likely under the model that included both the main effects and the interaction than under the model that included both main effects (logBF_10_ = 2.33).

We conducted separate *t*-tests for requested rule maintenance and requested rule switch trials to further parse the interaction, respectively. Conditional error probabilities of trials with a requested rule switch were lower with a requested response alternation (*M* = 0.22, *SE* = 0.03) than with an requested response repetition (*M* = 0.36, *SE* = 0.02), *t*(111) = −4.68, *p* < 0.001, Cohen’s *d* = −0.44, logBF_10_ = 7.52, which indicated the presence of a medium-sized ESE. On trials with a requested rule maintenance, we found no statistically significant difference between the trials with a requested response alternation (*M* = 0.16, *SE* = 0.02) and a requested response repetition (*M* = 0.18, *SE* = 0.02), *t*(111) = −1.16, *p* = 0.249, Cohen’s *d* = −0.11, logBF_10_ = −1.61. Thus, we did not find evidence for the presence of an EEE. Note that set-loss error probabilities on the requested-response-alternation sequences did not differ between the two types of possible errors on these trials, i.e., response repetitions (*M* = 0.07, *SE* = 0.01) or switches to a wrong response (*M* = 0.09, *SE* = 0.01), *t*(111) = −1.13, *p* = 0.259, Cohen’s *d* = −0.11, logBF_10_ = −1.63.

### 3.2. Exploratory Analyses

Exploratory analyses focused on the contrast between perseveration errors with a requested response repetition and a requested response alternation given the evidence for the presence of an ESE in the absence of an EEE, i.e., on the ESE. Exploratory correlation analyses revealed medium-sized correlations between the newly derived M-WCST conditional error probabilities and most subtests of the CERAD-NAB (see Table 4). Only one CERAD-NAB subtest (*Abbreviated Boston Naming Test*), and years of education, appeared to be more closely related to perseveration errors with a requested response repetition than to perseveration errors with a requested response alternation, but the opposite pattern never occurred. 

Figure 3 shows the ESE as a function of the diagnostic entities that comprised our sample of neurological patients (i.e., atypical Parkinson‘s disease (PD), early Alzheimer’s disease (AD)/mild cognitive impairment (MCI), frontotemporal lobar degeneration, vascular encephalopathy, stroke, multiple sclerosis, normal pressure hydrocephalus, depression, neuropathy, and absence of a neurological disease). There are several observations to note: 1) All of the patients who suffered from neurological diseases that affect the central nervous system and patients who suffered from depression showed considerable tendencies to commit perseveration errors. Patients who suffered from a neurological disease that affects the peripheral nervous system (neuropathy), and patients without a recognizable neurological (or psychiatric) condition, did not show pronounced tendencies to commit perseveration errors. 2) The ESE was observable within each diagnostic entity, which suggested that it represents a robust finding, which is solidly discernible, even in small samples (3 ≤ *n* ≤ 21). 3) As a corollary of the robustness of the ESE across diagnostic entities, a differential ESE was not easily detectable. In a purely descriptive sense, patients who suffered from neurodegenerative diseases that affect striatal (as in the case of atypical PD) and limbic (as in the case of early AD/MCI) circuits, as well as depressive patients, showed stronger ESE in comparison to patients who suffered from other neurological diseases that affect the central nervous system. However, an ANOVA with the factors group (atypical PD & early AD/MCI (*n* = 26) vs. frontotemporal lobar degeneration/vascular encephalopathy/stroke/multiple sclerosis/normal pressure hydrocephalus (*n* = 66)) and response sequence revealed the statistically significant main effect of response sequence, *F*(1,90) = 19.25, *p* < 0.001, $\eta_{p}^{2}$ = 0.18, but neither a statistically significant main effect of group, *F*(1,90) = 1.52, *p* = 0.22, $\eta_{p}^{2}$ = 0.02, nor a statistically significant interaction between the group and response sequence, *F*(1,90) = 2.71, *p* = 0.10, $\eta_{p}^{2}$ = 0.03. The Bayesian mixed model ANOVA indicated that the observed data were most likely under the model that included a main effect of response sequence (posterior probability = 0.566, logBF_10_ = 5.38), followed by the model that included the main effects of the response sequence and group (posterior probability = 0.252, logBF_10_ = 4.57), and by the model that included both main effects and the interaction between response sequence and group (posterior probability = 0.178, logBF_10_ = 4.22). The model that solely included the main effect of group was not superior to the null model (posterior probability = 0.001, logBF_10_ = −0.84). 

Taken together, the ESE did not seem to share a substantial portion of variance with any of the traditional neuropsychological measures of executive functioning. There was solely an association between the ESE and performance on the *Abbreviated Boston Naming Test*, but this singular finding might be attributable to chance (among the multitude of non-significant differential correlations). The ESE occurred in comparable strength in all of the patients who suffered from a neurological disease that affected the central nervous system, despite the impression that diseases affecting striatal (atypical PD) or limbic (early AD/MCI) circuits (as well as depression) might be associated with an enhancement of the ESE. However, the within-group inter-individual variability was obviously substantial, such that the apparent differences in the strength of the ESE failed to reach conventional levels of statistical significance in a reasonably powered comparison.

## 4. Discussion

The present WCST data that were obtained from a sample of neurological inpatients primarily revealed that a medium-sized error-suppression effect (ESE) occurred following negative feedback, whereas no error-enhancement effect (EEE) could be observed following positive feedback. Verbally expressed ‘incorrect’ feedback imposed a detectable reallocation of patient’s behavior, which occurred at the disadvantage of the most recently executed response. In contrast, ‘correct’ feedback did not entail a detectable reallocation of patient’s behavior into the direction of either an advantage or a disadvantage of the most recently executed response. These data suggest that negative verbal feedback selectively induces behavioral plasticity at the disadvantage of the recently executed responses on the WCST. Our findings seem to indicate that punishment-based avoidance learning [47,48] affects performance on the WCST. Our results illustrate how the novel measure of the ESE, which is based on a response-level of analysis, can enrich the repertoire of behavioral scores that can be derived from the WCST. 

The behavioral plasticity, which expresses itself in the ESE, is subject to what could perhaps be best referred to as a pre-cognitive task conception. That is, the meaningful units of behavior that are not abstract entities, such as color, shape, or number of the objects that are depicted on the WCST cards that are usually referred to as ‘task rules’. These entities are not immediately present as a ‘stimulus’ on the WCST cards, but they rather occur contingent on higher-level executive processes such as concept formation, abstraction, and rule inference. Here, the pure selection of reference cards that depict non-categorized stimuli is considered as the unit of analysis that deserves attention. Being viewed from a broader perspective, this response-related analysis represents a novel breakdown of WCST performance, and these response-specific mechanisms of behavioral control may take place in addition to—and independently from—executive control processes. The distinction between the response-related and executive processes is reminiscent of Luria’s [30] distinction between motor and cognitive perseveration.

It should be noted that the term ‘suppression’ serves merely as a descriptive term, which does not imply that some sort of inhibition induced the reallocation of patient’s behavior described above. Inhibition may play a role (putatively in the form of avoidance), but instead of that—or in addition to that—an enhancement of the alternative options for obtaining reinforcement may have occurred. For example, if an ‘incorrect’ feedback followed the selection of the leftmost card on a particular trial, this may have led to the inhibition (avoidance) of selecting that card, or, alternatively, to the enhancement of the alternative three cards.

With regard to the development of a comprehensive task theory of the WCST, the current data imply that a response-level of behavioral control should be considered, in addition to higher levels of executive control [28] (for a comprehensive overview, see [29]). More specifically, the selection of reference cards on the WCST seems to be modulated by punishment, in a manner that is independent of the presented stimulus card or the currently prevailing task rule. In comparison to these punishing effects of negative feedback, the reinforcing effects of positive feedback seem to be less effective in modulating reference-card selection on the WCST.

The exploratory analyses showed that the ESE did not share a substantial portion of variance with traditional neuropsychological measures of executive functioning, as assessed by the extended version of the CERAD-NAB. This finding seems to suggest that individual values in the ESE yield information that cannot be predicted from individual abilities on the more traditional measures of executive functioning. The exploratory analyses also pointed towards the direction that striatal (as it occurs in atypical PD patients) or limbic (as it occurs in early AD/MCI patients) circuit neuropathology may be associated with enhanced ESE in comparison to groups of patients who suffer from predominant cortical (as it occurs in frontotemporal lobar degeneration and stroke) or subcortical (as it occurs in vascular encephalopathy and multiple sclerosis) neuropathology. However, this post-hoc observation could not be substantiated with adequate statistical scrutiny. Further research that directly addresses this issue is required in order to reach firm conclusions regarding the neurological substrates of the ESE. The work that we presented in this study was less ambitious in that regard, because it merely served to establish the existence of an ESE rather than to elucidate its neurobiological underpinnings. 

The lack of a healthy group of participants is among the major limitations of the current study. It was solely based on clinical data that were obtained from currently hospitalized, brain damaged patients. As a consequence, we do not yet know whether the ESE solely exists in these patients, or alternatively, whether it occurs in similar strength, or in an attenuated manner, in healthy individuals. Another caveat lies in the fact that the number of trials on which the examined types of errors could occur was relatively low. Specifically, the number of trials with a requested rule switch, in combination with a requested response repetition, was quite low (Table 2). This issue needs careful consideration in future studies of the ESE, because low numbers of occasions on which an error of interest might occur limit the reliability of the measure. Future work on this issue should address this serious limitation, such that more occasions for the commitment of the errors of interest are provided, for example, by increasing the overall number of trials that are administered. However, the overall frequency of occurrence of positive and negative feedback was roughly balanced in our sample of patients (see Table 2), thereby ruling out the possibility that differences in surprise, hence, the salience, of the two different types of feedback might be responsible for the dissociation between error suppression in the case of negative feedback and error enhancement in the case of positive feedback.

The assessment of erroneous behavior has a long-standing tradition in clinical neuropsychology. In fact, the error counts are much more common than the response time measures, and this preference is partly due to the fact that brain-damaged patients are typically much more error-prone than healthy individuals. Thus, while the performance of healthy individuals remains, in most cases, error-free on assessments instruments, such as the Trail Making Test (e.g., [49]), many brain-damaged patients commit multiple errors during the performance of that task [50,51]. As a consequence, while the investigation of error proneness does not seem particularly promising for cognitive psychologists, the current data should encourage cognitive neuropsychologists to develop their assessment techniques further in that direction.

The WCST represents—despite of its numerous shortcomings (see Section 1. Introduction)—a benchmark instrument for the clinical assessment of executive functioning in individual patients. The current findings may lay the ground for the development of novel scores that may not be achieved otherwise. Specifically, the ESE on the WCST may be utilized as an indicator of the individual abilities of patients in lower-order response-related learning, which might be dissociable from executive control. The analysis of this indicator of response suppression on this classical neuropsychological test may facilitate WCST-based evaluation of dual process models of behavioral control. Such models repeatedly occur in diverse forms in the neuropsychological literature. Their common denominator is the distinction between lower (putatively subcortical) and higher (putatively cortical) levels of behavioral control. The contention scheduling vs. supervisory attentional model [52,53,54], and the distinction between habitual and goal-directed types of behavior are prominent examples of dual process models [55,56].

This is a behavioral investigation of WCST performance in a mixed neurological sample. We derived novel error metrics by stratifying the traditional set loss and perseverative errors. The separating rule set and response set allowed for the measurement of performance across four trial types, crossing rule set (i.e., maintain vs. switch) and response demand (i.e., repeat vs. alternate). Critically, these four trial types can be grouped based on trial-wise feedback on the *t* − 1 trials. Rewarded (correct) maintain *t* − 1 trials should lead to error enhancement when the response demands shift from repeat to alternate. In contrast, punished (incorrect) *t* − 1 trials should lead to error suppression when response demands shift from repeat to alternate. The results supported the error suppression prediction: An error suppression effect was observed across numerous patient samples. The error suppression rates did not correlate with performance on standard neuropsychological tests. We interpret these findings as providing evidence that WCST performance is multi-layered cognitive structure, which is driven by both rule and response sets (see Figure 4). Specifically, they suggest that pure response selection behavior, before categories are considered, is particularly susceptible to punishment-based learning (i.e. feedback on incorrect responses), at least in the population of neurological inpatients. However, the absence of a non-neurological control group is a limitation of our study, as there is no evidence that what was observed is related to the presence of neurological disease. Indeed, the absence of correlation with other neuropsychological measures might suggest the opposite—that punishment-based learning is a driver of WCST performance irrespective of the nature, or indeed the existence, of brain abnormality. Here, our main argument is that our observation is in and of itself a worthwhile finding, which invites conducting (confirmatory) future research with the ultimate goal of developing a comprehensive neuropsychological task theory of the WCST.

## 5. Conclusions

Our data point into the direction that performance on the WCST should be conceived as multi-layered, which implied that behavior on this classical task is putatively organized at multiple levels of control. In this study, we provided evidence for the effectuality of response-related learning on the WCST. Yet, response-related learning needs to be complemented by higher levels of behavioral control. Specifically, attentional plasticity seems to be one important contributor to successful performance on the WCST [22]. A comprehensive task theory of the WCST should envisage a multi-level architecture of WCST performance, which includes response-related and executive levels of behavioral control.

## Figures and Tables

**Figure 1 brainsci-09-00141-f001:**
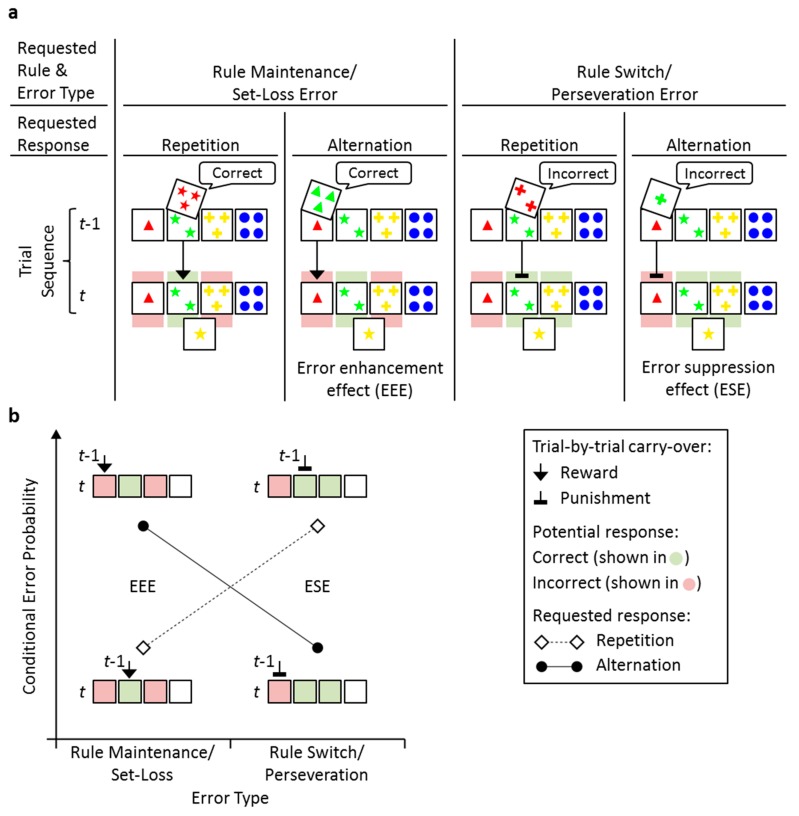
Exemplary sequences of two consecutive trials (i.e., *t* − 1, *t*), separately for requested rule maintenance and rule switch sequences (as indicated via positive and negative feedback on trial *t* − 1, respectively), and for requested response repetition/alternation sequences. Note that ‘odd’ errors on trial t were excluded from consideration (see Section 2. Materials and Methods for definition). (**a**). Left panels. ‘Correct’ feedback in response to sorts according to the Shape rule, which occur on trial *t* − 1, signal that the currently applied rule should be maintained on the upcoming trial; subsequent rule switches on trial *t* are hence erroneous because they obey either to the Number rule or to the Color rule. Note that these types of errors are traditionally considered as set-loss errors. Right panels. ‘Incorrect’ feedback in response to sorts according to the Number rule, which occur on trial *t* − 1, signal that a rule switch is requested on the upcoming trial (to obey either to the Shape rule or to the Color rule); maintenance of the currently applied rule on trial *t* would hence be erroneous. Note that these types of errors are traditionally considered as perseveration errors. The black symbols (arrows, cul-de-sac signs) illustrate potential response-specific carry-over from trial *t* − 1 to trial *t*, i.e., reward in case of ‘Correct’ feedback, or punishment, in case of ‘Incorrect’ feedback. (**b**). Further clarification of the response-level analysis of performance on the WCST (see Section 1. Introduction for details). Conditional Error Probability: Probability of an error given feedback type (‘correct’, ‘incorrect’) and response demand (repetition, alternation).

**Figure 2 brainsci-09-00141-f002:**
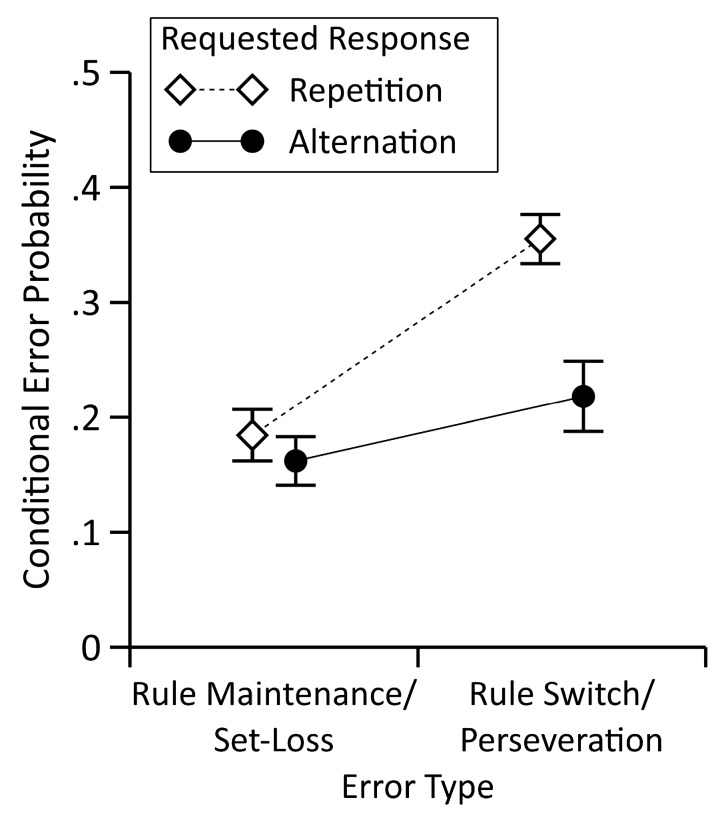
Mean conditional error probabilities, categorized into set-loss errors and perseveration errors, and separately for the four distinguishable rule/response sequences. See text for details. Error bars depict plus/minus one standard error of the mean. Conditional Error Probability: Probability of an error given feedback type (‘correct’, ‘incorrect’) and response demand (repetition, alternation).

**Figure 3 brainsci-09-00141-f003:**
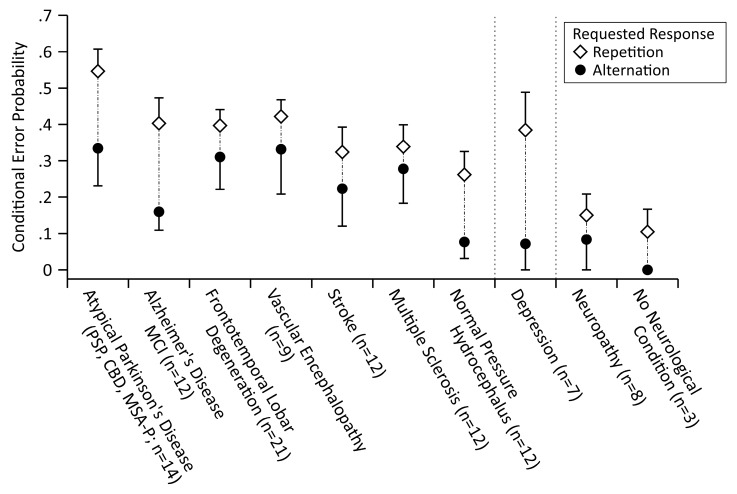
Mean conditional (perseveration) error probabilities in response to a requested rule switch (i.e., following negative feedback, which had been received on the previous trial), separately for the two types of response sequence (requested response repetition/alternation) and as a function of diagnostic entities (atypical Parkinson’s disease (progressive supranuclear palsy (PSP), cortico-basal degeneration (CBD), multi-system atrophy-Parkinsonian subtype (MSA-P)), (early) Alzheimer’s disease/mild cognitive impairment, frontotemporal lobar degeneration, vascular encephalopathy, stroke, multiple sclerosis, normal pressure hydrocephalus, depression, neuropathy, no neurological (or psychiatric) disease). For this analysis, *n* = 115 patients had sufficient data, but three patients who suffered from idiopathic Parkinson’s disease, a patient who suffered from dystonia, and a patient who suffered from multi-system atrophy-cerebellar subtype (MSA-C) were excluded from this analysis. An error-suppression effect was observed within each diagnostic entity. Error bars depict plus/minus one standard error of the mean. Conditional Error Probability: Probability of an error given negative feedback and response demand (repetition, alternation).

**Figure 4 brainsci-09-00141-f004:**
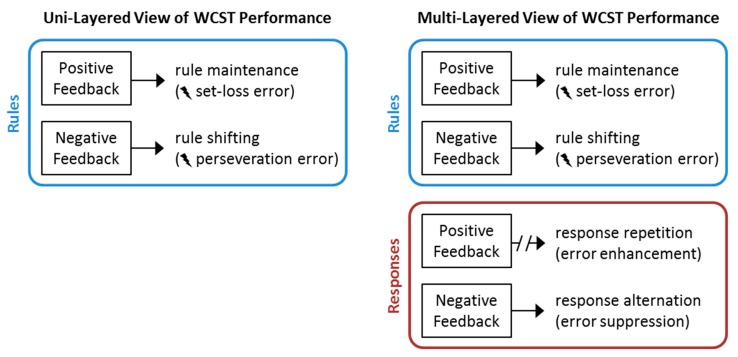
Summary of the factors that contribute to WCST performance. The traditional view of WCST performance considers a uni-layered structure governing task performance, such that positive feedback elicits rule maintenance and negative feedback elicits rule shifting. Failures of these cognitive processes are indicated by the occurrence of set-loss and perseveration errors, respectively. The extended view that emerges from this study suggests that a multi-layered structure governs WCST performance. Due to the multi-layered structure, the reception of feedback is associated with a credit assignment problem (e.g., [57]). Our findings suggest that negative feedback is associated with a modulation of the frequency of perseverations errors via response demands (observable as an error suppression effect). However, a modulation of the frequency of set-loss errors in response to positive feedback (i.e., an error enhancement effect) could not be discerned in the present study (as indicated by the broken arrow).

**Table 1 brainsci-09-00141-t001:** Sociodemographic and neuropsychological characteristics of included and excluded patients. See Table 3 for a detailed description of the Consortium to Establish a Registry for Alzheimer’s Disease Neuropsychological Assessment Battery (CERAD-NAB) scores.

	Included (*N* = 112)			Excluded (*N* = 34)		
	*M*	*SD*	*n*	*M*	*SD*	*n*
Age	61.08	12.26	112	63.71	12.43	34
education (years)	13.12	2.23	112	12.94	2.47	34
M-WCST categories	2.99	1.87	112	3.19	2.23	16 ^#^
CERAD-NAB (core)						
*Animal fluency*	−0.99	2.49	108	−1.50	1.38	34
*Abbreviated Boston Naming Test*	−0.11	1.26	109	−0.11	1.28	31
*Word list learning*	−1.31	1.39	112	−1.57	1.49	34
*Word list recall*	−0.95	1.27	112	−1.05	1.23	34
*Word list intrusions*	−0.62	1.30	112	−0.12	1.05	34
*Word list savings*	−0.85	2.00	110	−0.80	1.81	34
*Word list discriminability*	−0.62	1.27	112	−0.83	1.23	34
*Constructional praxis*	−0.89	1.24	111	−1.13	1.40	34
*Constructional praxis recall*	−1.16	1.54	111	−1.59	2.00	34
*Constructional praxis savings*	−0.54	1.18	111	−0.82	1.25	34
Adjusted Chandler score	73.12	12.00	108	66.78	12.40	31
CERAD-NAB (extension)						
*Letter fluency*	−1.12	1.30	108	−1.42	1.22	34
*Trail Making Test A*	−1.00	1.56	109	−1.30	1.57	32
*Trail Making Test B*	−1.14	1.47	101	−1.48	1.32	29

All CERAD-NAB scores are reported as standardized *z*-scores. *SD* = standard deviation; *n* = number of patients for which data on a specific variable are available. ^#^, This low number of patients is due to the exclusion of 18 patients who quit the execution of the M-WCST ahead of schedule.

**Table 2 brainsci-09-00141-t002:** Mean number of trials on which occasions for the types of errors of interest occurred, mean number of errors, and mean conditional error probabilities.

Requested Rule & Error Type	Rule Maintenance/Set-Loss Error		Rule Switch/Perseveration Error	
Requested Response	Repetition	Alternation	Repetition	Alternation
Errors Committed	1.25 (0.14)	1.33 (0.15)	8.55 (0.67)	0.76 (0.12)
Errors Possible	8.27 (0.31)	12.25 (0.53)	20.00 (0.78)	2.63 (0.15)
Conditional Error Probability	0.18 (0.02)	0.16 (0.02)	0.36 (0.02)	0.22 (0.03)

Standard error of the mean is shown in parentheses.

**Table 3 brainsci-09-00141-t003:** Subtests scores of the extended CERAD-NAB.

Subtest	Short Description
CERAD-NAB (core) *Animal fluency*	Number of animals within 1 minute
*Abbreviated Boston Naming Test*	Number of correctly named objects (max. = 15)
*Word list learning*	Immediate recall of 10 words, three repetitions (max. = 30)
*Word list recall*	Delayed recall of 10 words (max. = 10)
*Word list intrusions*	Number of erroneously recalled words on all *Word list learning* and *Word list recall* trials
*Word list savings*	$=\;\frac{\mathrm{Word}\;\mathrm{list}\;\mathrm{recall}}{\mathrm{Word}\;\mathrm{list}\;\mathrm{learning},\;\mathrm{third}\;\mathrm{trial}}\times 100$
*Word list discriminability*	Recognition of 10 words on a 20 words list (includes 10 distractors) $=\left\{1-\frac{\left(10-\mathrm{Hits}\right)+\left(10-\mathrm{Correct}\;\mathrm{Rejections}\right)}{20}\right\}\times 10$ (max. = 10)
*Constructional praxis*	Copying of four geometrical forms (max. = 11)
*Constructional praxis recall*	Delayed recall of four geometrical forms (max. = 11)
*Constructional praxis savings* CERAD-NAB (extension)	$=\;\frac{\mathrm{Constructional}\;\mathrm{praxis}\;\mathrm{recall}}{\mathrm{Constructional}\;\mathrm{praxis}}\times 100$
*Letter fluency*	Number of words beginning with letter ‘s’ within 1 minute
*Trail Making Test A* *Trail Making Test B*	Time needed in seconds (max. = 180 by truncation) Time needed in seconds (max. = 300 by truncation)

The *Mini Mental State Examination* was not assessed. The *Trial Making Test B/A* quotient was not considered.

**Table 4 brainsci-09-00141-t004:** Associations between Modified Wisconsin Card Sorting Test (M-WCST) conditional error probabilities and extended CERAD-NAB subtest scores.

		Spearman Correlation Coefficients (*r*_s_) for Perseveration Errors		Difference in Correlation Coefficients (*z*) ^Δ^
		Requested Response		
	*n*	Repetition	Alternation	
CERAD-NAB (core)				
*Animal fluency*	108	−0.34 *	−0.40 *	0.63
*Abbreviated Boston Naming Test*	109	−0.32 *	−0.09	−2.30 *
*Word list learning*	112	−0.34 *	−0.29 *	−0.61
*Word list recall*	112	−0.34 *	−0.29 *	−0.54
*Word list intrusions*	112	0.19 *	0.23 *	−0.39
*Word list savings*	110	−0.24 *	−0.22 *	−0.23
*Word list discriminability*	112	−0.22 *	−0.14	−0.75
*Constructional praxis*	111	−0.39 *	−0.27 *	−1.32
*Constructional praxis recall*	111	−0.46 *	−0.43 *	−0.42
*Constructional praxis savings*	111	−0.31 *	−0.32 *	0.11
Adjusted Chandler score	108	−0.44 *	−0.42 *	−0.29
CERAD-NAB (extension)				
*Letter fluency*	108	−0.36 *	−0.41 *	0.54
*Trail Making Test A*	109	0.50 *	0.38 *	1.40
*Trail Making Test B*	101	0.55 *	0.46 *	0.97
Age	112	0.32 *	0.34 *	−0.17
Years of education	112	−0.34 *	−0.14 *	−2.09 *

M-WCST = Modified Wisconsin Card Sorting Test, CERAD-NAB = Consortium to Establish a Registry for Alzheimer’s Disease Neuropsychological Assessment Battery. **^Δ^**, Correlation coefficients were compared according to the procedure outlined by Meng et al. [46]. The correlation between the two types of M-WCST errors, needed to compare their respective correlations with CERAD-NAB scores, was *r*_s_ = 0.46. *, *p* < 0.05 (two-sided).

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
