# Peer review of "Multiple Levels of Control Processes for Wisconsin Card Sorts: An Observational Study"

_brainsci, 2019, doi:10.3390/brainsci9060141_

Round 1
Reviewer 1 Report
The authors present a novel and theoretically relevant approach to the scoring of errors on the Modified Wisconsin Card Sorting test (M-WCST) by stratifying traditional set loss errors and perseverative errors in successive t-1 and t trials. This resulted in a 2 x 2 task design consisting of two factors that potentially influence the commission of errors: high-order rule set (maintain vs switch) and low-order response demand (repeat vs alternate). The resulting four error types were grouped based on the correct or incorrect feedback on t-1 trials indicating high-order rule set maintenance or switching, respectively. According to this the authors could rightly identify two potentially relevant response tendencies, either an error enhancement tendency or an error suppression tendency in either case, when response demands shift from repeat to alternate. This new exploratory analysis of errors was applied to M-WCST data from a respectable sample of N=112 neurological patients. The results showed an error suppression effect (ESE) in patients showing many distinct neurological pathologies --but no presence of any error enhancement effect (EEE), which were interpreted as evidence that ESE provide an index of response-related avoidance learning on the WCST.
In the Introduction section the authors rightly raise several limitations underlying the scoring of WCST performance, and they highlight the need of a solid theoretical framework for interpreting the different types of WCST errors. Based on current theoretical models in cognitive neuroscience, the authors convincingly argue for the existence of multiple, hierarchically organized levels of control processes to account for, i.e., WCST perseverative errors, which thus could be attributed either to high-level cognitive perseveration, or to low-level motor perseveration. Accordingly, the authors here propose a new M-WCST scoring method in order to classify traditional WCST perseverative errors and set-loss errors not only in the light of categorical learning of abstract WCST rules, but also according to low-level pre-categorical learning of specific motor responses from trial t-1 to trial t. The description of this new scoring system, as well as the specific hypotheses raised for potential response-related outcomes under rule maintenance and rule shifting conditions are adequately illustrated and clarified with an explicit figure and the accompanying figure legend. In particular, two explicit predictions were made regarding (a) a set-loss "error enhancement effect" (EEE) for motor response alternations at trial t under rule maintenance t-1 conditions, and (b) a perseverative "error suppression effect" (ESE) for motor response alternations at trial t under rule switching t-1 conditions. All in all, the rationale of the study, the logic of the experimental approach, and the new behavioral measures used are all quite well explained in the Introduction and Method sections.
In the results section both frequentist and Bayesian statistics are used in order to quantify evidence in favor of the null hypothesis under each model, in compliance with published and widely accepted standards. Likewise, both confirmatory and exploratory analyses were conducted to fully explore EEE and ESE effects and their association with various neurological conditions. The results showed an error suppression effect (ESE) in patients showing many distinct neurological pathologies --but no presence of any error enhancement effect (EEE). These results were interpreted as evidence that ESE provide an index of response-related avoidance learning on the WCST. All in all, results are well organized and quite easy to follow, and possible limitations are discussed. The discussion is well balanced and the conclusions about the presence of only an error suppression effect (ESE) seem generally supported by the data. In sum, I think that this is a solid paper for publication in Brain Sciences, and the manuscript fits well into the profile of the journal.
However, I have a few minor points the authors may want to address in a revised version off the manuscript:
1. One issue that potentially limits the significance and ultimate interpretation of results is the absence of a control group of healthy participants without any neurological damage. Certainly, this would warrantee that the absence of an error enhancement effect (EEE) of response alternation under rule maintenance conditions is not the consequence of neurological damage per se.
2. (p. 6) Can the authors expand on the definition of “odd” errors? Maybe these were not related to any categorical dimension? What was the proportion of these "odd” errors relative to the other four error types? This is relevant to understand to what extent set loss errors could be determined by chance alone.
3. (p. 6) Table 2 should also express the conditional probabilities of each error type, given that this seemed to be the quantity that was analyzed and represented in Figure 2.
4. (p. 6) Table 2 shows a clear unbalance in the mean probabilities of different types of errors in this Modified WCST. The question is to what extent this structural bias in the a priori probability of each type of error could determine the final outcome.
5. (p. 9) Table 4. Please clarify the meaning of the fourth column: Comparisons (z)
6. The main conclusion of this study is that rule maintenance t-1 conditions don’t seem to have a great impact on response-dependent trial t errors, whereas rule switching t-1 conditions do have an impact on response-dependent trial t errors. However, previous WCST studies have also shown that an unexpected change on trial-wise feedback cues may have a detrimental effect on the behavioral response to the subsequent WCST trial in the sequence (Kopp & Lange, 2013 Front Hum Neurosci). Since the present M-WCST had a much lower probability of displaying negative feedback cues compared to positive feedback cues, how could this sensory (or probabilistic) aspect of task performance have influenced the present ESE effects?
Author Response
Reviewer 1 | ||
comment | response | change in the manuscript (page numbers refer to pages in the revised version of the manuscript) |
1. One issue that potentially limits the significance and ultimate interpretation of results is the absence of a control group of healthy participants without any neurological damage. Certainly, this would warrantee that the absence of an error enhancement effect (EEE) of response alternation under rule maintenance conditions is not the consequence of neurological damage per se. | This is indeed a limitation of our work that we explicitly acknowledge in our submitted version of the manuscript (see Discussion).
“However, the absence of a non-neurological control group is a limitation of our study, as there is no evidence that what was observed is related to the presence of neurological disease. Indeed, the absence of correlation with other neuropsychological measures might suggest the opposite - punishment-based learning is a driver of WCST performance irrespective of the nature, or indeed the existence, of brain abnormality. Our main argument here is that our observation is in and of itself a worthwhile finding, which invites conducting (confirmatory) future research with the ultimate goal of developing a comprehensive neuropsychological task theory of the WCST.” | No further changes to the manuscript. |
2. (p. 6) Can the authors expand on the definition of “odd” errors? Maybe these were not related to any categorical dimension? What was the proportion of these "odd” errors relative to the other four error types? This is relevant to understand to what extent set loss errors could be determined by chance alone. | We included a definition of ‘odd’ errors and report the overall occurrence of ‘odd’ errors. Participants in the final sample committed 106 ‘odd’ errors in total, which is 0.02% of all trials. Set-loss errors occurred on approximately 17% of all trials following a positive feedback. Thus, it seems unlikely that set-loss errors are determined by chance. | Page 6, lines 211-214: “We excluded the rarely occurring ‘odd’ errors on trial t from consideration (0.02% of all trials). An ‘odd’ error was committed by selecting the stimulus card that matches no valid task rule (e.g., the far right stimulus card on trial t in Figure 1a).” |
3. (p. 6) Table 2 should also express the conditional probabilities of each error type, given that this seemed to be the quantity that was analyzed and represented in Figure 2. | We included conditional error probabilities in Table 2. | Table 2. |
4. (p. 6) Table 2 shows a clear unbalance in the mean probabilities of different types of errors in this Modified WCST. The question is to what extent this structural bias in the a priori probability of each type of error could determine the final outcome. | You are right that the different types of errors differed with regard to the frequency of their occurrence. However, our main outcome measure is conditional probability of the error types in question. Thus, an occurrence of 10 errors on 20 opportunities to commit that error would yield an identical value compared to 2 errors on 4 opportunities (p = 0.5 in both cases). Having said this, the sole difference between these two measures might be related to the reliability that they offer (with the former’s, based on more observations, exceeding the latter’s reliability). However, we do not want to touch this (complicated) psychometric issue in the current manuscript. | No changes to the manuscript. |
5. (p. 9) Table 4. Please clarify the meaning of the fourth column: Comparisons (z) | This was indeed not sufficiently clear. We added the missing information to Table 4. | Table 4. |
6. The main conclusion of this study is that rule maintenance t-1 conditions don’t seem to have a great impact on response-dependent trial t errors, whereas rule switching t-1 conditions do have an impact on response-dependent trial t errors. However, previous WCST studies have also shown that an unexpected change on trial-wise feedback cues may have a detrimental effect on the behavioral response to the subsequent WCST trial in the sequence (Kopp & Lange, 2013 Front Hum Neurosci). Since the present M-WCST had a much lower probability of displaying negative feedback cues compared to positive feedback cues, how could this sensory (or probabilistic) aspect of task performance have influenced the present ESE effects? | Kopp & Lange’s (2013) paper addressed a sequential effect in WCST performance. It was focused on conditions for rule inference, in particular sequences like t-2 = incorrect, t-1 = incorrect (+ rule change), t = correct rule identified? Thus, that paper addressed high-level cognitive abilities, i.e., rule inference. Here, we were interested in low level conditioning-like processes, i.e. attribution of feedback information not only at the rule level (‘incorrect’ tells me to abandon the recently executed rule), but also – implicitly – at the response level (‘incorrect’ punishes the recently executed response). See also our response to your fourth comment for an answer to the mentioned probabilistic aspects of task performance. Please note that the overall frequency of occurrence of positive and negative feedback was roughly balanced in our sample of patients (see Table 2), thereby ruling out the possibility that differences in surprise, hence salience, of the two different types of feedback might be responsible for the dissociation between error suppression in case of negative feedback and error enhancement in case of positive feedback. | Page 12, lines 445-449: “However, the overall frequency of occurrence of positive and negative feedback was roughly balanced in our sample of patients (see Table 2), thereby ruling out the possibility that differences in surprise, hence salience, of the two different types of feedback might be responsible for the dissociation between error suppression in case of negative feedback and error enhancement in case of positive feedback.” |

Reviewer 2 Report
The authors present a novel and interesting scoring form of the Modified Wisconsin Card Sorting test in a wide variety of neurological disorders. Patients were also evaluated with a relatively comprehensive neuropsychological battery. Introduction is adequate. Statistical analysis seems to be correct. And discussion is also adequate.
The main limitations of the study is the absence of a control group, the very heterogeneous sample, and the fact that patients were evaluated during hospitalization. Some of these limitations are acknowledged by the authors and included in the manuscript.
I have only some minor corrections:
-2.1. Participants. Specific neurological diagnosis should be added here (they are only reported in Figure 3).
-Maybe the authors could consider to add a Figure summarizing the main factors associated to WCST performance according to the findings of this investigation and previous ones.
-In my opinion, the discussion about "striatal vs limbic vs cortical" is not very accurate according to current models in neuropsychology. Furthermore, atypical parkinsonisms also showed cortical involvement (frontal in PSP, parietal and frontal in CBD), stroke is very heterogeneous, etc.
Author Response
Reviewer 2 | ||
comment | response | change in the manuscript (page numbers refer to pages in the revised version of the manuscript) |
2.1. Participants. Specific neurological diagnosis should be added here (they are only reported in Figure 3). | We added specific neurological diagnoses of the complete sample. | Page 4, lines 154-160: “Diagnostic assignments were conducted by an experienced neurologist (TS) who was blinded for the neuropsychological characteristics of individual patients with regard to the effects of interest. Diagnostic assignments were atypical Parkinson’s disease (progressive supranuclear palsy (PSP), cortico-basal degeneration (CBD), multi-system atrophy-Parkinsonian subtype (MSA-P); n = 25), (early) Alzheimer’s disease/mild cognitive impairment (n = 14), frontotemporal lobar degeneration (n = 26), vascular encephalopathy (n = 10), stroke (n = 13), multiple sclerosis (n = 14), normal pressure hydrocephalus (n = 14), depression (n = 13), neuropathy (n = 10), cognitive impairment with unknown origin (n = 4), and no neurological (or psychiatric) disease (n = 3).” |
Maybe the authors could consider to add a Figure summarizing the main factors associated to WCST performance according to the findings of this investigation and previous ones. | We added a figure that summarizes the main factors associated with WCST performance, with the traditional view depicted on the left-hand side and the extended (based on the results that were obtained in our study) view on the right-hand side. | We added Figure 4 on page 13.
See also page 14, lines 492-494: “We interpret these findings as providing evidence that WCST performance is governed by a multi-layered cognitive structure, driven by both rule and response sets (see Figure 4)” |
In my opinion, the discussion about "striatal vs limbic vs cortical" is not very accurate according to current models in neuropsychology. Furthermore, atypical parkinsonisms also showed cortical involvement (frontal in PSP, parietal and frontal in CBD), stroke is very heterogeneous, etc. | Your criticism at this point is right. At this point in time, we simply do have no clues about potential neural correlates of the error suppression effect. With regard to that discussion, we merely intend to nominate some of the major candidates, which however, are not independent neural systems, in particular with regard to the cortical – striatal distinction. We highlighted throughout our manuscript, that the main purpose of the current study should be viewed in the detection of the error suppression effect, be it modulated by the presence of brain disease (and if so, by which kind of brain disease) or not. | No changes to the manuscript. |
Reviewer 3 Report
The authors present a novel error metric of the Modified Wisconsin Card Sorting Test (M-WCST) by separating rule and response set. This is a very interesting additional perspective on performance from a cognitive neuroscientific view. Whether these make their way into neuropsychological practice with common WCST variants is questionable. More details are following below.
Introduction:
The WCST was developed to study “shift of set” and is thus used to assess cognitive flexibility or set shifting. In the introduction, a detailed description of this cognitive dimension is missing. The authors more generally rely on overall executive functions with an emphasis on “control processes exerted at an abstract, categorical level”.
Participants
More than 23% of the patients were excluded from analysis. This is a rather high exclusion rate and as reasoned later on page 7, “We had to exclude 18 patients from analysis who were unable to complete the M-WCST. We further excluded 16 patients from analysis who did not have the opportunity to commit all of the above defined types of error.”
This clearly limits the general applicability and novel analytic approach. Please specify in more detail why the patients did not have the “opportunity” to commit all of the defined error types. Did the ordering of the trials differ between individuals so that not all error types were possible in each subject? Or didn’t they make any mistakes, or didn’t they attain a sufficient number of positive feedback trials followed by a mistake?
Material and Design:
Why did the authors use a fixed order of M-WCST task rules throughout their study? Shouldn’t this limit the measurement of cognitive flexibility, especially in patients with preserved cognitive abilities, as categories became predictable?
The so-called “efficient” error on page 5, is this the same as the “odd” error on page 6?
The frequency distribution of “Errors Committed” and “Errors Possible” across the four conditions in Table 2 explains my introductory comment above. These error frequencies make a systematic analysis for a novel metric difficult. Consequently, for statistical analysis, the authors had to compute proportional error scores. Why did not they simply divide “Errors Committed” by “Errors Possible”? This would also ease the understanding of the P(e) values for example on Figure 2.
Page 7, line 218: Please replace “CERAD-NAB” with “CERAB-NAB”
Statistical Analyses
As many readers are not so familiar with Bayesian ANOVA presented in the Results part: Could you shortly explain, for example, what logBF10 = 2.33 means, i.e. why it indicates “that the observed data were ten times more likely…” (line 278)?
Author Response
Reviewer 3 | ||
comment | response | change in the manuscript (page numbers refer to pages in the revised version of the manuscript) |
Introduction | ||
The WCST was developed to study “shift of set” and is thus used to assess cognitive flexibility or set shifting. In the introduction, a detailed description of this cognitive dimension is missing. The authors more generally rely on overall executive functions with an emphasis on “control processes exerted at an abstract, categorical level”. | We completely agree. There are several aspects of your criticism: The notion of control is a very loose description, but what on the other hand does “set shifting” actually mean? Given the vagueness of all these terms, the current manuscript gains only limited precision by introducing “set shifting”, in particular in light of the fact that “set shifting” was not in our focus. Our group had several earlier publications that directly attempted to treat “set shifting”, and its neural correlates, as assessed by the WCST. We hence addressed your comment in the following way: | Page 2, lines 58-60: “The WCST is perhaps the most frequently used neuropsychological test for the assessment of cognitive flexibility, an ability that involves shifting between cognitive sets as required by task demands [14, 17–18].”
|
Participants | ||
More than 23% of the patients were excluded from analysis. This is a rather high exclusion rate and as reasoned later on page 7, “We had to exclude 18 patients from analysis who were unable to complete the M-WCST. We further excluded 16 patients from analysis who did not have the opportunity to commit all of the above defined types of error.” This clearly limits the general applicability and novel analytic approach. Please specify in more detail why the patients did not have the “opportunity” to commit all of the defined error types. Did the ordering of the trials differ between individuals so that not all error types were possible in each subject? Or didn’t they make any mistakes, or didn’t they attain a sufficient number of positive feedback trials followed by a mistake? | Error propensity is in general an aspect of behavior that is not under the control of the experimenter/examiner. We explained that if for example a patient committed perseveration errors on each single trial (believe it or not, it happens in some patients), this patient had no opportunity to commit a set loss error simply because he/she would never receive a positive feedback (see page 7, lines 240-243). This is a general problem for all of us who study errors as they occur on any behavioral paradigm. So, we excluded patients (like the one in that example) when they had no chance to commit one of the error types under consideration. With regard to the overall proportion of patients that had to be excluded based on overall performance (i.e., failure to complete the WCST), we felt that it may be of importance to report that some patients (in any sample) will fail on the WCST simply because it is a relatively complex psychological task that lies beyond the cognitive capabilities of some individuals (see page 7, lines 237-238). | No changes to the manuscript. |
Material and Design | ||
Why did the authors use a fixed order of M-WCST task rules throughout their study? Shouldn’t this limit the measurement of cognitive flexibility, especially in patients with preserved cognitive abilities, as categories became predictable? | This choice is primarily a matter of personal preference, which has its basis in clinical experience. Fixed ordering of task rules enforces patients to start with the color rule, and it is that feature of fixed ordering that - in our opinion – provides its competitive advantage. Many patients are enforced – at the very beginning of the examination – to think about potential task rules, whereas other regimes tend to be associated with postponed thinking about potential task rules. Moreover, given the low number of task runs on the M-WCST, it is unlikely that participants can apply acquired knowledge about the rule sequence during later stages of the task. Even if they could use rule-sequence knowledge in that way, it would only help them to directly switch to the correct rule when switching rules – an aspect of task performance that is not analyzed in the present contribution. | No changes to the manuscript. |
The so-called “efficient” error on page 5, is this the same as the “odd” error on page 6? | In order to clarify the distinction between error types, we included a detailed definition of ‘odd’ errors. | Page 6, lines 212-214: “An ‘odd’ error was committed by selecting the stimulus card that matches no valid task rule (e.g., the far right stimulus card on trial t in Figure 1a).” |
The frequency distribution of “Errors Committed” and “Errors Possible” across the four conditions in Table 2 explains my introductory comment above. These error frequencies make a systematic analysis for a novel metric difficult. Consequently, for statistical analysis, the authors had to compute proportional error scores. Why did not they simply divide “Errors Committed” by “Errors Possible”? This would also ease the understanding of the P(e) values for example on Figure 2. | We agree. Conditional error probabilities were computed by dividing the number of committed errors by the number of possible errors. We changed the explanation of the computation of conditional error probabilities and changed the wording throughout the manuscript. Note that we also changed the notation P(e) in Figures 1 to 3 to “conditional error probability”. | Page 6, lines 210-211: “Conditional error probabilities were computed by dividing the number of committed errors by the sum of committed errors and correct responses.” |
Page 7, line 218: Please replace “CERAD-NAB” with “CERAB-NAB” | We corrected the misspelling accordingly. | Page 7, line 231. |
Statistical Analyses | ||
As many readers are not so familiar with Bayesian ANOVA presented in the Results part: Could you shortly explain, for example, what logBF10 = 2.33 means, i.e. why it indicates “that the observed data were ten times more likely…” (line 278)? | We included a description of logBF followed by an example of the interpretation of logBF. | Page 7, lines 254-258: “In addition to posterior probabilities, we reported logarithmized Bayes factors (logBF), which quantify the support for a hypothesis over another. For example, logBF10 = 2.30 indicates that the alternative hypothesis is approximately ten times more likely than the null hypothesis, as the corresponding Bayes factor is exp(logBF10) = 10.” |
Round 2
Reviewer 3 Report
No more comments